# Expression of Low Level of VPS35-mCherry Fusion Protein Diminishes Vps35 Depletion Induced Neuron Terminal Differentiation Deficits and Neurodegenerative Pathology, and Prevents Neonatal Death

**DOI:** 10.3390/ijms22168394

**Published:** 2021-08-04

**Authors:** Yang Zhao, Fulei Tang, Daehoon Lee, Wen-Cheng Xiong

**Affiliations:** 1Department of Neurosciences, School of Medicine, Case Western Reserve University, Cleveland, OH 44106, USA; zhaoy8092@126.com (Y.Z.); dxl660@case.edu (D.L.); 2Department of Neuroscience and Regenerative Medicine, Medical College of Georgia, Augusta University, Augusta, GA 30912, USA; FTANG@augusta.edu

**Keywords:** Vps35, retromer complex, neurodegeneration, gliosis

## Abstract

Vps35 (vacuolar protein sorting 35) is a key component of retromer that consists of Vps35, Vps26, and Vps29 trimers, and sortin nexin dimers. Dysfunctional Vps35/retromer is believed to be a risk factor for development of various neurodegenerative diseases. *Vps35^Neurod6^* mice, which selectively knock out Vps35 in Neurod6-Cre+ pyramidal neurons, exhibit age-dependent impairments in terminal differentiation of dendrites and axons of cortical and hippocampal neurons, neuro-degenerative pathology (i.e., increases in P62 and Tdp43 (TAR DNA-binding protein 43) proteins, cell death, and reactive gliosis), and neonatal death. The relationships among these phenotypes and the underlying mechanisms remain largely unclear. Here, we provide evidence that expression of low level of VPS35-mCherry fusion protein in *Vps35^Neurod6^* mice could diminish the phenotypes in an age-dependent manner. Specifically, we have generated a conditional transgenic mouse line, *LSL-Vps35-mCherry*, which expresses VPS35-mCherry fusion protein in a Cre-dependent manner. Crossing *LSL-Vps35-mCherry* with *Vps35^Neurod6^* to obtain *TgVPS35-mCherry*, *Vps35^Neurod6^* mice prevent the neonatal death and diminish the dendritic morphogenesis deficit and gliosis at the neonatal, but not the adult age. Further studies revealed that the *Vps35-mCherry* transgene expression was low, and the level of *Vps35* mRNA comprised only ~5–7% of the *Vps35* mRNA of control mice. Such low level of VPS35-mCherry could restore the amount of other retromer components (Vps26a and Vps29) at the neonatal age (P14). Importantly, the neurodegenerative pathology presented in the survived adult *TgVps35-mCherry*; *Vps35^Neurod6^* mice. These results demonstrate the sufficiency of low level of VPS35-mCherry fusion protein to diminish the phenotypes in *Vps35^Neurod6^* mice at the neonatal age, verifying a key role of neuronal Vps35 in stabilizing retromer complex proteins, and supporting the view for Vps35 as a potential therapeutic target for neurodegenerative diseases.

## 1. Introduction

Retromer complex, a key endosomal protein sorting machinery, recognizes specific membrane cargo proteins that are concentrated in discrete regions of the endosomal membranes, and retrieves cargo proteins to the trans-Golgi network or the plasma membrane [1,2,3]. These functions are achieved by two sub-protein complexes of the retromer: One for the cargo-selection, and the other for membrane deformation. The cargo recognition retromer sub-protein complex consists of the vacuolar protein sorting (Vps) subunits, Vps26, Vps29, and Vps35. The membrane deformation sub-complex consists of sortin nexin (SNX) dimers [4,5]. A growing list of retromer cargos has been identified, which includes cation independent mannose 6-phosphate receptors (CI-MRP) [6,7], the Wnt transport protein Wntless/MIG-14 [8,9], sortilin or sortilin-related receptor (SorL1 or SorLA) [10,11,12,13], amyloid precursor protein (APP) [14,15], BACE1 [16,17], TREM2 [18,19], RANK [20]. Interestingly, many of these cargos are involved in the pathogenesis of neurodegenerative diseases. For example, while SorLA is a risk gene for late onset AD, sortilin is implicated in frontotemporal dementia (FTD), an early onset dementia that is characterized by atrophy of the frontal and/or temporal lobes of the cortex [21,22]. Thus, retromer is implicated in regulation of the development of neurodegenerative disorders.

Indeed, Vps35, a key component of retromer cargo recognition sub-complex, has been reported as an active player in the pathogenesis of neurodegenerative diseases [2,23,24]. Several lines of evidence suggest that loss of the Vps35 function contributes to the development of neurodegenerative diseases. First, Vps35 is decreased in postmortem hippocampus of patients with AD [25]. Second, deficiency of Vps35 in mice induces AD-, PD-, and FTD-relevant pathologies [16,26,27]. For example, genetic reduction of Vps35 in mouse models increases the production of β-amyloid peptide (Aβ) (a key pathological factor of AD), synaptic dysfunction, impairs mitochondrial dynamics and function, and induces memory and cognitive deficits [16,26,28,29]. Third, the gain of function of Vps35 in a triple transgenic AD mouse model, 3xTg-AD, which harbors a human mutant *PSEN1* (M146V) knock-in, and mutant *APP* (KM670/671NL) and *MAPT* (P301L) transgenes, can fully reduce the AD-relevant phenotypes [24,30]. Notice that in this study, AAV-Vps35 (the Vps35 expression is under the control of synpasin-1 promoter) was bilaterally injected into the cerebral ventricles of the newborns of 3xTg-AD mice, and the AD-relevant brain phenotypes in 12 months-old-mice are diminished [24]. Whereas, these observations support the view for loss of the Vps35 function to contribute to the AD development, and the increase of the neuronal Vps35 expression at the neonatal age could attenuate AD-relevant pathology in the aged 3xTg-AD mouse model, the details of Vps35 therapy (e.g., dose, critical time window, and key brain regions that AAV-Vps35 are expressed) and the underlying mechanisms of the Vps35 function remain elusive.

Vps35 is highly expressed in developing pyramidal neurons of mouse neocortex and hippocampus, and its expression level is peaked at the neonatal stage (P10–P15) [16,17]. Loss of Vps35 in developing pyramidal neurons results in dendritic morphogenesis and maturation defects and axonal spheroid formation in pyramidal neurons [17,27]. Mice that selectively deplete *Vps35* gene in embryonic (by *Neurod6-Cre*), but not postnatal (by *CamkII-Cre*), pyramidal neurons display not only neuronal terminal differentiation deficits, but also neurodegenerative pathology, such as elevation of cortical neuronal death, accumulations of Tdp43 (TAR DNA-binding protein 43), P62/Sqstm1, and reactive glial pathology [27]. These studies suggest that Vps35 expression in mouse developing pyramidal neurons is required for their terminal differentiation as well as prevention of neurodegeneration, revealing a coupling between neuron terminal differentiation deficit with the neurodegeneration. However, the exact relationship among these deficits remains largely unclear.

Here, we provide evidence that expression of VPS35-mCherry fusion protein in *Vps35^Neurod6^* mice prevents their neonatal death, and attenuates the dendritic morphogenesis deficit and gliosis, but does not increase in P62 and Tdp43, at the neonatal age. However, in the adult age, the neurodegenerative pathology, including dystrophic neurites and gliosis, occurred in *Vps35^Neurod6^* mice expressing VPS35-mCherry fusion protein. Further examination showed that the Vps35-mCherry transcript expression was low (~5–7% as compared with that of control mice), and such a low level of VPS35-mCherry was able to restore the amount of other retromer components (Vps26a and Vps29) at the neonatal age (P14), but not the adult age. Therefore, the neurodegenerative pathology appeared in the survived young adult *TgVps35-mCherry; Vps35^Neurod6^* mice. Together, these results suggest the sufficiency of low level of VPS35-mCherry fusion protein to diminish the deficit in neuronal terminal differentiation, cell death, and gliosis at the neonatal age, and thus to prevent neonatal death, demonstrating the neuronal Vps35 function in stabilizing retromer complex proteins in an age dependent manner, and supporting the view for Vps35 as a potential therapeutic target for neurodegenerative diseases.

## 2. Results

### 2.1. Generation of Conditional Transgenic Mice That Express VPS35-mCherry Fusion Protein in a Cre-Dependent Manner

To investigate the Vps35 function in suppression of the development of neurodegenerative diseases, we generated *LSL-Vps35-mCherry* mice, which express VPS35-mCherry fusion protein in a Cre-dependent manner, since there is an insertion of a loxp-flanked “stop” sequence between the CAG promoter and *Vps35-mCherry* transgene (Appendix A). As expected, the *Vps35-mCherry* transgene was only detectable when Cre recombinase was co-expressed, but not in the absence of Cre, in HEK293 cells (Appendix A), demonstrating its Cre-dependence. Then, we crossed the *LSL-Vps35-mCherry* mice with *Neurod6-Cre* mice to generate *LSL-Vps35-mCherry*; *Neurod6-Cre* mice (referred to as *TgVps35^Neurod6^*) to determine if the transgene is expressed in vivo in a Cre-dependent manner. Note that *Neurod6-Cre* expresses Cre largely in post-mitotic cortical and hippocampal pyramidal neurons of neocortex starting at ~E13.5 [31]. As shown in Appendix A, *TgVps35^Neurod6^* mice had normal body weight and brain structure compared with the control mice (Appendix A). Consistent with results in HEK293 cells, we only detected the expression of VPS35-mCherry protein in cortex and hippocampus, not in cerebellum where Cre is not expressed (Appendix A). The immunostaining analysis showed VPS35-mCherry protein expression in *TgVps35^Neurod6^* neocortex (layers II-III, layers IV-V) and hippocampus (CA1-3) (Appendix A). The co-immunostaining analysis using antibodies against mCherry and Camk2a, Aldolase C and Parvalbumin (PV), markers for pyramidal neurons, astrocytes and interneurons, respectively, showed that VPS35-mCherry protein was co-distributed with Camk2a, but not Aldolase C^+^ astrocytes nor PV^+^ interneurons (Appendix A). These results confirmed VPS35-mCherry protein’s specific expression in excitatory pyramidal neurons, demonstrating its Cre-dependence in vivo. Subcellularly, we found that the VPS35-mCherry protein was partially co-localized with Vps26a, GM130 (a marker for Golgi), and Rab7 (a marker for late endosomes) (Appendix A), suggesting that this VPS35-mCherry fusion protein partially, not fully, mimics the endogenous Vps35 distribution pattern.

### 2.2. Preventing Vps35^Neurod6^ Mice from Neonatal Death by Expressing VPS35-mCherry Fusion Protein

Previous research showed that *Vps35^Neurod6^* mice lagged in postnatal development including cessation of weight gain after 1 week and neonatal lethality before P27 for reasons that have not yet been determined [27]. We crossed the *TgVps35^Neurod6^* mice with *Vps35^f/f^* mice to generate *Vps35^f/f^* (referred to as control), *Vps35^f/f^; Neurod6* (referred to as *Vps35^Neurod6^*), and *LSL-Vps35-mCherry; Vps35^Neurod6^* mice (referred to as *TgVps35^Neurod6^; KO*) (Figure 1A). The *TgVps35^Neurod6^; KO* mice had a comparable body weight to that of control mice at ages of postnatal day 7 to 14 (P7–P14) (Figure 1B,C). Remarkably, whereas *Vps35^Neurod6^* mice had undergone neonatal death (all die before P28), *TgVps35^Neurod6^; KO* mice survived into adult ages, up to more than 8 months old (Figure 1D). Consistent with body weight, the brain size in *TgVps35^Neurod6^; KO* mice was comparable with that of control mice (Figure 1E,F). This view was further supported by the Nissl-staining analysis of brain sections, which showed comparable cortical thickness in *TgVps35^Neurod6^; KO* mice with that of control mice at P14 (Figure 1G,H).

### 2.3. Decrease of Cell Death and Restoring of Terminal Differentiation of Neuronal Dendrites in Vps35^Neurod6^ Pups by Expressing VPS35-mCherry Fusion Protein

Accompanied with the reduced cortical thickness in *Vps35^Neurod6^* mice was the increase of caspase-3^+^ cell death [27,32]. Therefore, we evaluated the effect of *Vps35-mCherry* transgene on the cell death by immunostaining analysis of the active caspase-3 positive apoptotic cells in control and mutant neocortex. As shown in Figure 2A,B, an increase in active caspase-3^+^ cell density was detected in P14 *Vps35^Neurod6^* mice, in line with our previous report [27]. As expected, a significant reduction in caspase-3^+^ cell density was observed in *TgVps35^Neurod6^; KO* mice (Figure 2A,B), suggesting that the expression of VPS35-mCherry fusion protein could attenuate this phenotype. Furthermore, we examined neurodegeneration in *TgVps35^Neurod6^; KO* mice, and compared it with that in *Vps35^Neurod6^* mice, by the use of Jade C staining in control, *Vps35^Neurod6^* and *TgVps35^Neurod6^; KO* mice at P14 (Figure 2C). While Jade C positive signals were detected in both *Vps35^Neurod6^* and *TgVps35^Neurod6^; KO* brains in L2-3 regions (Figure 2C,D), they were significantly lower in *TgVps35^Neurod6^; KO* cortex than that of *Vps35^Neurod6^* mice (Figure 2C,D), suggesting a partial rescue of the neurodegeneration phenotype by the *Vps35-mCherry* transgene. Next, we evaluated the effect of VPS35-mCherry on dendritic maturation defect in *Vps35^Neurod6^* mice, a deficit tightly coupled with the neurodegenerative phenotype [27]. Using Golgi staining analysis of P14 control, *Vps35^Neurod6^* and *TgVps35^Neurod6^; KO* cortex to view neuronal dendritic morphologies, we observed disrupted dendritic morphologies, including reduced dendritic complexity and length, largely at layers 2–3 neurons in *Vps35^Neurod6^* cortex (Figure 2E–G), consistent with the previous report [27]. Remarkably, this deficit was largely abolished in *TgVps35^Neurod6^; KO* mice (Figure 2E–G), suggesting a restoration of dendritic morphogenesis and maturation in *Vps35^Neurod6^* mice by expressing VPS35-mCherry fusion protein.

### 2.4. Attenuation of Gliosis in Vps35^Neurod6^ Pups by Expressing VPS35-mCherry Fusion Protein

The defective dendritic morphogenesis is often associated with glial activation, which was the case in *Vps35^Neurod6^* mice [27]. Therefore, we further examined glial cell responses in *TgVps35^Neurod6^; KO* mice, compared with that of control and *Vps35^Neurod6^* mice. As shown in Figure 3, the P14 *Vps35^Neurod6^* neonatal cortex exhibited increases in Iba1^+^ microglial cell intensity per cell, as well as elevations of GFAP^+^ astrocytes. However, both the intensities of Iba1^+^ microglia and GFAP^+^ astrocytes were not increased in *TgVps35^Neurod6^; KO* cortex, which were at comparable levels as those in control mice (Figure 3A,B), suggesting that a complete inhibition of glial activation in *TgVps35^Neurod6^; KO* neonatal cortex, supporting the view for the association of glial responses with neurodegeneration. Similar to the results in the cortex, an increase of the intensity of Iba1^+^ microglia were detected in hippocampus of *Vps35^Neurod6^* mice, which was not observed in the age- and gender matched *Vps35^Neurod6^* mice expressing VPS35-mCherry fusion protein at P14 (Figure 3C,D).

### 2.5. Partial Reduction of P62 and Tdp43 Proteins in TgVps35^Neurod6^; KO Pubs

In addition to the neuronal terminal differentiation deficits and gliosis in *Vps35^Neurod6^* mice, we also detected increases of P62/sequestosome 1 (SQSTM1), one factor that targets specific cargoes for autophagy, and Tdp-43, a DNA-RNA binding protein, in *Vps35^Neurod6^* cortical neurons (Figure 4) [33,34,35,36], both proteins often associated with the pathology of FTD [21,37]. To determine if expressing VPS35-mCherry fusion protein could reduce this brain pathology in *Vps35^Neurod6^* mice, we performed both coimmunostaining and Western blot analyses. Immunostaining analysis showed a partial reduction of P62 in the neocortex L2-3 of *TgVps35^Neurod6^; KO* mice as compared with that in *Vps35^Neurod6^* mice (Figure 4A,B). Western blot analysis showed higher protein levels of P62 and Tdp43 in insoluble fractions of homogenates of both *Vps35^Neurod6^* and *TgVps35^Neurod6^; KO* cortex than those in control mice. However, both P62 and Tdp43 proteins were lower in *TgVps35^Neurod6^; KO* than those in *Vps35^Neurod6^* mice (Figure 4C,D). These results suggested that the aggregates of FTD-associated proteins in *TgVps35^Neurod6^*; *KO* mice were partially reduced.

### 2.6. Neurodegenerative Pathology in Adult TgVps35^Neurod6^; KO Mice

Next, we examined phenotypes in different aged *TgVps35^Neurod6^; KO* mice. First, we examined the neuronal morphology in the survived *TgVps35^Neurod6^; KO* mice at age of 2-months-old (MO), compared with the age- and gender- matched control mice (*Vps35^f/f^*) (Figure 5A,B). We examined the neuronal morphology by stereotactic injection of AAV-Syn-EGFP to fluorescently label neurons of cortex in control and *TgVps35^Neurod6^; KO* mice (Figure 5A–D). The quantification analysis showed marked reductions in dendritic intersections and length in the mutant mice (Figure 5C,D), implicating the neurodegenerative phenotype. Then, we examined other neurodegeneration associated features, including increases in P62 and gliosis. Indeed, the P62 proteins were increased in insoluble fractions of homogenates of *TgVps35^Neurod6^; KO* neocortex (Figure 5E,F); and the GFAP^+^ astrocytes and Iba1^+^ microglia were also elevated in *TgVps35^Neurod6^; KO* neocortex (Figure 5G–I). Note that the level of Tdp43 was comparable with that of control mice (Figure 5E,F). Together, these results support the view for the existence of neurodegenerative pathology in adult *TgVps35^Neurod6^; KO* mice.

Second, we wondered whether these neuron morphological and degenerative phenotypes could be detected in neonatal ages (but after P14). To this end, we examined the dendritic morphology of neurons in hippocampus CA1 and entorhinal cortex of control (*Neurod6 Cre*), *Vps35^Neurod6^* and *TgVps35^Neurod6^; KO* brains by stereotactic injection of AAV-Syn-mCherry to fluorescently label neuronal morphology (injected at P22 and examined at P29) (Appendix A). As shown in Appendix A, in hippocampus CA1, the mCherry labelled pyramidal neuron’s dendrites had a reduced dendritic branching as compared with that of control mice. However, a significant improvement was noted as compared with that in *Vps35^Neurod6^* mice (Appendix A). The Sholl analysis showed a lower number of dendritic intersections in *TgVps35^Neurod6^; KO* mice (Appendix A). In addition, quantification of the apical and basal dendritic length indicated a reduction of dendritic process in *TgVps35^Neurod6^; KO* neurons, as compared with those in control mice (Appendix A). In entorhinal cortex of *TgVps35^Neurod6^; KO* mice, a comparable dendritic complexity and length were detected as compared with those in control mice (Appendix A). These deficits were similar, but not identical, to the phenotypes observed by Golgi staining analysis (Figure 2E), suggesting a partial rescue of the dendritic length and complexity at this neonatal age.

Third, we examined other phenotypes, including cell death, cortical thickness, and gliosis in control, *Vps35^Neurod6^*, and *TgVps35^Neurod6^; KO* mice at age of P21. The P21 *TgVps35^Neurod6^; KO* cortex showed more active caspase-3 positive apoptotic cells and Jade C^+^ signals than those of P14 *TgVps35^Neurod6^; KO* mice (Figure 6A–D). Accompanied with the increased cell death and neurodegeneration were a partial reduction of cortical thickness in *TgVps35^Neurod6^; KO* mice (Figure 6E,F), and appearance of the glial responses (Figure 6G,I), although these phenotypes had some improvement compared with those of *Vps35^Neurod6^* mice (Figure 6G–J).

In aggregates, these results demonstrate an age-dependency of the VPS35-mCherry effect, revealing the neurodegenerative pathology (altered dendritic morphology, increased P62 and reactive gliosis) in the survived young adult *TgVps35^Neurod6^; KO* mice.

### 2.7. Age-Dependent Restoration of Retromer Complex in Vps35^Neurod6^ Mice Expressing VPS35-mCherry Fusion Protein

Vps35, a key component of retromer, interacts with Vps26a and Vps29 and is critical for the protein complex stability [4,38]. Therefore, we examined Vps26a and Vps29 levels in control, *Vps35^Neurod6^*, and *TgVps35^Neurod6^; KO* mice at various ages (P7, P14, and P21). As shown in Figure 7, Western blot analysis detected abundant Vps35, Vps26a, and Vps29 protein levels in control cortex, and they were largely reduced or undetectable in *Vps35^Neurod6^* mice, in line with the view for the Vps35 function in stabilizing the complex. Interestingly, expressing the VPS35-mCherry fusion protein was capable of restoring Vps26a and Vps29 in P14 *TgVps35^Neurod6^; KO* brain (both cortex and hippocampus), at comparable levels with those of control mice (Figure 7C,D), suggesting a full rescue. However, at ages of P7 and P21, the levels of Vps26a and Vps29 were higher in *TgVps35^Neurod6^; KO* brain than those of *Vps35^Neurod6^* mice, but remained lower as compared with those of control mice (Figure 7A,B,E,F), indicating a partial rescue. Notice that the *VPS35*-mCherry protein was detectable by antibody against mCherry, which showed a molecule weight of 110 kDa, but not by anti-Vps35 antibody (Figure 7), suggesting the lower sensitivity of the Vps35 antibody and/or the low level of VPS35-mCherry fusion protein expression. Together, these results suggest an age dependent restoration of the retromer complex protein in *Vps35^Neurod6^* mice expressing VPS35-mCherry fusion protein, correlating with its age-dependent rescue effects on the neurodegenerative pathology.

### 2.8. Low Level of Vps35-mCherry’s Expression

Furthermore, we examined *Vps35-mCherry*’s mRNA levels in *TgVps35^Neurod6^* and *TgVps35^Neurod6^; KO* mice by quantitative real-time RT-PCR (qRT-PCR) analysis. Primers for *Vps35-mCherry* and endogenous *Vps35* were generated as indicated in Figure 8A. Using primers for *Vps35-mCherry*, *Vps35-mCherry*’s mRNAs were detected in neocortex and hippocampus of *TgVps35^Neurod6^* and *TgVps35^Neurod6^; KO* mice, but not in control and *Vps35^Neurod6^* mice, suggesting the specificity (Figure 8B). However, using primers for endogenous *Vps35*, the mRNA levels in both *Vps35^Neurod6^* and *TgVps35^Neurod6^; KO* brains were lower than those in control and *TgVps35^Neurod6^* mice. However, the *Vps35* mRNA levels were slightly higher in *TgVps35^Neurod6^; KO* than in *Vps35^Neurod6^* mice (Figure 8B). Further quantification analysis showed that the exogenous *Vps35-mCherry*’s mRNAs were at levels of about 4–6% over the total *Vps35* mRNA (Figure 8B), suggesting a low level of *Vps35-mCherry*’s expression. To verify this view, we performed the RNA-scope analysis, a high resolution In Situ method. As shown in Figure 8C, the Vps35 signal was detected in the neocortex of control, *Vps35^Neurod6^* and *TgVps35^Neurod6^; KO* mice. Consistent with the result of qRT-PCR, the signal was markedly reduced in *Vps35^Neurod6^* mice, with a slight increase in *TgVps35^Neurod6^; KO* brain, as compared with that of *Vps35^Neurod6^* mice (Figure 8C,D). Moreover, we examined *Vps26a* and *Vps29* mRNA levels in control, *Vps35^Neurod6^* and *TgVps35^Neurod6^; KO* neocortex and hippocampus. No significant change was observed between the control and *Vps35^Neurod6^* mice, as well as the *TgVps35^Neurod6^; KO* mice at P14 (Figure 8E), suggesting the Vps35 effect on the protein stability of Vps26a and Vps29, but not their transcription. Together, these results suggest a low level of VPS35-mCherry expression, and such a low level of VPS35-mCherry fusion protein could diminish most of the deficits detected in P14 *Vps35^Neurod6^* mice, a critical window for their survival.

## 3. Discussion

The retromer complex is a highly conserved multimeric protein complex and has been linked to cargo retrieval from endosomes to the trans-Golgi network [39,40,41]. Vps35/retromer dysfunction is involved in pathogenesis of multiple neurodegenerative diseases, including AD and PD, FTD, and ALS [16,26,27,42]. Although numerous studies have demonstrated that Vps35 is an active player and directly involved in development of neurodegenerative diseases, the underlying mechanisms remain largely unclear. In this study, we expressed Vps35 in pyramidal neuron by crossing *LSL-Vps35-mCherry* mice with *Vps35^Neurod6^* mice, which develop terminal differentiation deficits and neuro-degenerative pathology. Expressing a low level of VPS35-mCherry fusion protein could prolong the life span of *Vps35^Neurod6^* mice up to 8 months or longer, and partially rescued the bodyweight of *Vps35^Neurod6^* mice. Further analysis showed that the neuronal dendritic differentiation deficits, cell death, and reactive glial responses in *Vps35^Neurod6^* mice were largely diminished at neonatal age (P14), but not a young adult age. Finally, the protein and mRNA levels of Vps35-mCherry in *TgVps35^Neurod6^; KO* mice appeared to be as low as ~5–7% of the *Vps35* mRNAs of control mice. Such a low level of expression of this fusion protein is sufficient to restore the levels of other retromer components (Vps26a and Vps29), attenuate the neurodegenerative deficits at the neonatal age (P14), prolong the life span, but not prevent the neurodegenerative pathology in older ages.

Genetically manipulated mouse models have proven to be valuable tools for our understanding of pathological mechanisms of human diseases. Here, we used a conditional transgenic mouse line, with a knock-in Cre-regulated expression of cDNA constructs with a loxp-stop-loxp cassette insertion between the CAG promoter and *Vps35-mCherry* cDNA. This system enables approaches for controlled expression of VPS35-mCherry constructs. The diminished brain pathology in *Vps35^Neurod6^* mice was detected at the neonatal, not adult age, by expressing the VPS35-mCherry fusion protein. These observations suggest that the rescue by VPS35-mCherry fusion protein appears to be “transient” or age-dependent. The rescue by VPS35-mCherry may be also dose dependent. Notice that the expression of *VPS35-mCherry* transgene is regulated not only by CAG promoter, but also depends on Cre-recombinase. The Neuro6-promotor drive Cre expression may be peaked at P14, which results in higher activity of VPS35-mCherry and better rescue effect. In addition, the low level of *Vps35-mCherry* transgene expression may be due to the random insertion of the *LSL-Vps35-mCherry* transgene cassette.

It is noteworthy that there was a coupling of neuronal terminal differentiation deficit with the glial activation, but not with the P62/TDP43 levels (Figure 2, Figure 3 and Figure 4). Activation of glial cells are implicated in the pathogenesis of many neurodegenerative disorders [43,44,45]. Activation of glial cells, including microglia and astrocytes, in response to changes in neurons, such as neuron injury and increase of Aβ, may further contribute to the neuron/tissue damage. The accumulations of Tdp43 and P62/SQSTM1 are FTD-like neuropathology, and neurons with Tdp43 aggregates suggest a combination of loss of nuclear function and gain of toxic function. Our results that glial cell responses were nearly abolished, while the increased P62 and Tdp43 remained, in P14 *TgVps35^Neurod6^; KO* mice (Figure 4) suggest that the changes in P62 and Tdp43 may not be direct inducers for the glial responses. However, the observations of the tight association between defective dendritic maturation with the glial activation in these mutant mouse lines at age of P14 (Figure 2E–G and Figure 3) suggest that they may have interdependent effects under the influence of Vps35-loss in pyramidal neurons.

It is known that P62 accumulates with ubiquitin-containing aggregated [46,47]. Our results of a partial reduction of Tdp43 and P62 levels in *TgVps35^Neurod6^; KO* mice at neonatal and young adult ages (Figure 4 and Figure 6E–H) suggest that low doses of Vps35 expression may not be enough to remedy the accumulations of Tdp43 and P62 protein levels. In line with the levels of Tdp43 and P62 proteins, the cell death and Jade C^+^ neurodegeneration were also remained in P14 *TgVps35^Neurod6^; KO* mice (Figure 2A–D), suggesting an association of these pathological events. Additionally noteworthy is that the *TgVps35^Neurod6^*; *KO* mice can survive up to 8 months or more, whereas the *Vps35^Neurod6^* mice undergo neonatal death before P27. While these results suggest an importance for mouse survival after passing the development of the neonatal critical time window, the exact reasons for neonatal death remain to be determined. Remarkably, the low level of VPS35-mCherry fusion protein expression in *TgVps35^Neurod6^; KO* mice was sufficient to prevent the neonatal death phenotype (Figure 1D). These results lead us to believe that the expression of low level of Vps35 might be a useful approach for the therapy of patients with neurodegenerative diseases who may have dysfunctional retromer. It is of interest to further investigate this view in future studies.

## 4. Materials and Methods

### 4.1. Animals

The LSL-Vps35-mCherry transgenic mice were generated by use of the pCCALL2 plasmid. The Vps35-mCherry sequence was cloned into the downstream of the Loxp-STOP-Loxp sites in pCCALL2 plasmid. The construct was linearized using ScaI/SfiI and electroporated into ES cells. Positive ES cell clones were injected into C57BL6/J blastocysts. Successful expression of Vps35-mCherry was verified by PCR, Western blot, and mCherry fluorescence. The Vps35 floxed (*Vps35^f/f^*) mice were generated, maintained, and genotyped as described previously [27,48]. The *NeuroD6-Cre* (also called *Nex-Cre*) mice were kindly provided by Klaus-Armin Nave [31]. The *Vps35^Neurod6^* mice were generated by breeding *Vps35^f/f^* mice with *Neurod6-Cre* mice. The *TgVps35^Neurod6^* mice were generated by crossing *LSL-Vps35-mCherry* with *Neurod6-Cre* mice. The *TgVps35^Neurod6^; KO* mice were generated by crossing *LSL-Vps35-mCherry* mice with *Vps35^Neurod6^* mice. All phenotypic characterizations of *Vps35^Neurod6^* mice, *LSL-Vps35-mCherry* mice, *TgVps35^Neurod6^* mice, *TgVps35^Neurod6^; KO* mice and their control mice were in C57BL/6 background for >6 generations.

All the mouse lines were confirmed by genotyping analysis with PCR. Genotyping of mouse lines was performed by PCR of tail prep DNAs. The following primer pairs were used: *Vps35* flox- 5’ AACCAGCTCCCAACAAAATG 3’ and 5’ GCTTGGTCCCACTCACATTT 3’ to amplify a 180-bp in *WT* allele and a 220-bp in *floxed* allele; *Neurod6-Cre*- 5’ GAGTCC TGGAATCAGTCTTTTTC 3’ and 5’ CCGCATAACCAGTGAAACAG 3’ to amplify a 550-bp in *Cre* allele; *LSL-Vps35-mCherry*- 5’ ACACATTGGAGCACTTGCGCTCAAGACG 3’ and 5’ CTGCCCTTCGCCTGGGACATCCTGT 3’ to amplify a 315-bp in *mCherry* allele.

All the mice were maintained on a 12-h light-dark cycle with ad libitum access to water and food. Male mice were used for all the studies. All the experiments with animals were performed with the approval of the Institutional Animal Care and Use Committee of Case Western Reserve University (CWRU) according to the National Institute of Health (NIH) guidelines (protocol no. 2017-0121, approved on 11 May 2020).

### 4.2. Antibodies for Immunostaining and Western Blotting

The following antibodies were used: Rabbit polyclonal anti-Vps35 antibody was generated against the murine Vps35 C-terminal sequence as described previously [16] (diluted 1:1000); rat monoclonal anti-mCherry (diluted 1:500; Thermo Fisher Scientific, Waltham, MA, USA; M11217); mouse monoclonal anti-β-Actin (diluted 1:1000; Cell Signaling, Danvers, MA, USA; 3700S); mouse monoclonal anti-NeuN (diluted 1:1000; EMD Millipore, Billerica, MA, USA; MAB377); mouse monoclonal anti-Camk2a (diluted 1:300; Cell Signaling, Danvers, MA, USA; 50049); goat polyclonal anti-Aldolase C (diluted 1:200; Santa Cruz, Dallas, TX, USA; sc-12065); rat monoclonal anti-PV (diluted 1:500; abcam, Cambridge, MA, USA; ab27853); rabbit polyclonal anti-Vps26a (diluted 1:500; abcam, Cambridge, MA, USA, 23892); mouse monoclonal anti-GM130 (diluted 1:500; BD, San Jose, CA, USA, 610822); mouse monoclonal anti-Rab7 (diluted 1:500; abcam, Cambridge, MA, USA, ab50533); goat polyclonal Iba1 (diluted 1:500; abcam, Cambridge, MA, USA, ab5076); rabbit monoclonal anti-GFAP (diluted 1:200; Cell Signaling, Danvers, MA, USA; 12389); rabbit polyclonal anti-cleaved caspase-3 (diluted 1:300; Cell Signaling, Danvers, MA, USA; 9661); mouse monoclonal anti-P62 (diluted 1:500; abcam, Cambridge, MA, USA; ab56416); rabbit polyclonal anti-TDP43 (C-terminal) (diluted 1:500; Proteintech, Chicago, IL, USA; 12892–1-AP); mouse monoclonal anti-Vps29 (diluted 1:200; Santa Cruz, Dallas, TX, USA; sc-398874).

### 4.3. Nissl and Golgi Staining

For Nissl staining, sections of mouse brains were incubated in 0.1% cresyl violet solution for 5–10 min, then quickly washed in distilled water and dehydrated in increasing ethanol concentrations (70%, 80%, 90%, and 100% ethanol for 2–3 min step by step). The sections were cleared with xylene for 2–5 min, mounted with DPX Mountant (Sigma-Aldrich, St Louis, MO, USA; 44581) for histology and dried in fume hood. The cortical thickness was measured in Nissl-stained sections using the ImageJ software. Golgi staining was performed using the FD Rapid GolgiStain Kit (FD NeuroTechnologies, Columbia, MD, USA) as described previously [49]. For the neuronal morphometric analysis, pyramidal neurons were randomly selected from somatosensory cortical layer II/III. Images were acquired using a BZX microscope at 10× or 40× magnification, and the total dendritic length and Sholl analysis were calculated by Neuron J (ImageJ v1.8.0, NIH).

### 4.4. Adeno-Associated Virus (AAV) Injection

For cortex injection, the surgical procedure was performed as described previously with a slight modification [50]. Briefly, mice were anesthetized with Ketamine/Xylazine (HENRY SCHEIN, Melville, NY, USA; #056344) and the head was fixed in a stereotaxic device. After antiseptic treatment, the skull was exposed and cleaned using 1% H_2_O_2_. Then, holes were drilled into the skull and viruses (pAAV-hSyn-EGFP Addgene, Boston, MA, USA; #50465) were bilaterally injected into the cortex. After injection, mice were daily cared for 5 days. Two weeks later, mice were anesthetized and perfused and the brains were dissociated for immunofluorescence analysis.

For hippocampus CA1 and entorhinal cortex injection, the surgical procedure was performed as described previously [51]. After anesthetization and fixation as well as treatment of the mice with 1% H_2_O_2_ as described above, stereotactic surgery was performed to deliver a volume of 1 μL (1 × 10^12^) of pAAV-synP-FLEX-splitTVA-EGFP-B19G (Addgene, Boston, MA, USA; #52473) [52] into the right side of the CA1 or entorhinal cortex.Then, the rabies virus RbV-EnvA-ΔRgp-MCh (1 μL, 0.5 × 10^12^) was delivered into the same locations 6 or 7 days later. The motorized stereotaxic injector (Stoelting, Kiel, WI, USA; 53311) was used to infuse the virus into the hippocampus at a rate of 0.1 μL/min.

### 4.5. Tissue Processing and Immunofluorescence

Mice with indicated genotypes were anesthetized and perfused transcardially with 4% paraformaldehyde (PFA; pH 7.4). The brains were dissociated and postfixed overnight, then mounted in agarose prior to vibratome sectioning. In addition, 35–50 µm free-floating vibratome sections of the brains were collected in PBS, then blocked and permeabilized with a blocking solution containing 10% BSA and 0.5% Triton X-100 in PBS for 1 h at room temperature and incubated with primary antibodies in a blocking solution overnight at 4 °C. The brain sections were then washed with a PBS buffer and incubated with the appropriate secondary antibodies at 1:1000 (Thermo Fisher Scientific, Waltham, MA, USA; Alexa Fluor conjugates) for 2 h at room temperature. Brain sections were washed and incubated with DAPI (Thermo Fisher Scientific, Waltham, MA, USA) to reveal the cell nuclei. After washed with PBS, mouse the brain sections were mounted for confocal imaging with the Zeiss LSM 800 system. Images were also taken with the BZX fluorescence microscope.

### 4.6. Western Blotting

Tissues were dissected and homogenized in a lysis buffer (20 mM Tris-HCl (pH 7.4), 150 mM NaCl, 1% NP-40, 0.5% Triton X-100, 1 mM phenylmethylsulfonyl fluoride (PMSF), 1 mM EDTA, 5 mM sodium fluoride, and 2 mM sodium orthovanadate, and supplemented with a protease inhibitor cocktail (Roche, Indianapolis, IN, USA)). After a 20 min of incubation on ice, frozen and thawed twice, and ultracentrifuged at 100,000× *g*, 4 °C for 30 min, we got the supernatant as the soluble fraction. Then, the Triton-insoluble pellets were sonicated with 2× volume per original sample weight in urea-lysis buffer (7 M urea, 2 M thiourea, 4% CHAPS, and 30 mM Tris, pH 7.5, and a protease inhibitor cocktail (Roche)) and centrifuged at 100,000× *g* for 30 min at 22 °C, and we got the supernatants as the insoluble fraction [27]. The protein concentration was measured by the BCA assay (Pierce Biotechnology, Rockford, IL, USA) and the supernatants were mixed with a loading buffer. Then, for Western blot analysis, based on the protein molecular weight, we used 8% or 10% SDS-PAGE gels to separate the proteins and transferred them onto Nitrocellulose Blotting Membranes. After blocking with 5% low fat dried milk (in 1x TBST) for 1 h at room temperature, antigen specific primary antibodies were diluted to incubate the membrane overnight at 4 °C, then species-specific horseradish-peroxidase-conjugated secondary antibodies (1:5000, Thermo Fisher Scientific, Waltham, MA, USA) and ECL kit (Pierce Biotechnology, Rockford, IL, USA) were used for visualizing the target proteins. For quantitative analysis, the Image J software were used to analyze the protein bands, which were normalized to the loading control (β-actin).

### 4.7. Jade C Staining

Jade C staining was performed as described previously [27]. Fluoro-Jade C is a polyanionic fluorescein derivative that can sensitively and selectively bind to degenerative neurons [53]. Brain sections were mounted on gelatin-coated slides and subjected to Jade C staining according to the manufacturer’s protocol (Fluoro-Jade C, Millipore, Billerica, MA, USA). Then, sections were imaged with the Zeiss LSM 800 system v2.3 lite.

### 4.8. Quantitative Real-Time RT-PCR (qRT-PCR)

Total RNA was extracted from the cortex and hippocampus using a TRIzol reagent (Life Technologies, Waltham, MA, USA) as previously described [54]. Then, purified RNA (5 µg) was used for cDNA synthesis with the GoScript Reverse Transcription System (Promega, Madison, WI). The cDNA products were subjected for subsequent qPCR using SYBR green (QIAGEN, Germantown, MD, USA) in the CFX96 real-time system (Bio-Rad, Hercules, CA, USA). The primers used were as follows:Vps35-mCherry, forward: 5′ AGAGTCTGAGGGGCCAATCT 3′,reverse: 5′ CCTTGAAGCGCATGAACTCC 3′;Vps35, forward: 5′ TTGGTAGAAATGTGCCGTGGTGT 3′,reverse: 5′ CATCCGCACCCAGAGCTTATTCA 3′;Vps26a, forward: 5′ CTCGTGCTTGTTGATGAGGAGG 3′,reverse: 5′ CGCTGGTGAAAGTTCGTCCTCT 3′;Vps29, forward: 5′ TGCACCAAGGAGAGCTACGACT 3′,reverse: 5′ ACTTGGTGTCCGTGGATCAGAC 3′;GAPDH, forward: 5′ AAG GTC ATC CCA GAG CTG AA 3′,reverse: 5′ CTG CTT CAC CAC CTT CTT GA 3′.

Each sample was repeated at least 3 times, and the mRNA level was normalized to GAPDH using the 2^−^^△△Ct^ method.

### 4.9. RNA Scope

The RNA scope was performed in the mouse brain using the RNAscope^®^ Multiplex Fluorescent Detection Kit (Noble Park North, Australia; PN323110). The Vps35 probe and hybrid oven were purchased from the ACDbio company, and all the procedures were performed according to the manufacturer’s protocol. Briefly, the mice were deeply anesthetized and perfused with 4% paraformaldehyde (PFA; pH 7.4). Next, the brains were dissected and post-fixed with 4% PFA for 3 h and dehydrated with 30% sucrose in PBS. Then, the brains were cryo-sectioned into 10 µm sections using a freezing microtome (Leica, Buffalo Grove, IL, USA). Subsequent processing followed the RNAscope^®^ Multiplex Fluorescent Reagent Kit v2 User Manual. Finally, sections were imaged with the Zeiss LSM 800 system.

## Figures and Tables

**Figure 1 ijms-22-08394-f001:**
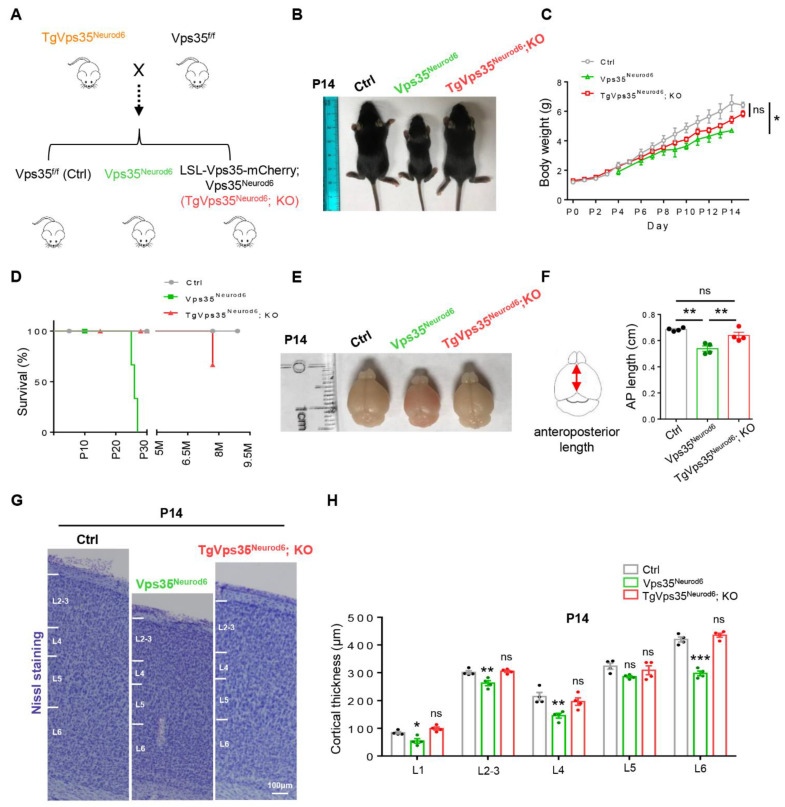
Generation of *TgVps35^Neurod6^; KO* mice. (**A**) *TgVps35^Neurod6^* mice were crossed with *Vps35^f/f^* mice to generate *TgVps35^Neurod6^; KO* mice. (**B**) *TgVps35^Neurod6^; KO* mice appeared indistinguishable from control littermates at P14. (**C**,**D**) Body weight and Kaplan-Meier survival curves of *TgVps35^Neurod6^; KO* mice, *Vps35^Neurod6^* mice and littermate controls. (**E**) Representative images showing the brain size of control, *Vps35^Neurod6^* and *TgVps35^Neurod6^; KO* mice at P14. (**F**) Quantification of the brain size (*n* = 4 mice per group; one-way ANOVA with Tukey’s multiple-comparison test). (**G**) Representative Nissl staining of control, *Vps35^Neurod6^* and *TgVps35^Neurod6^; KO* mice at P14. (**H**) Quantification analysis of Nissl staining that revealed a comparable cortical thickness of *TgVps35^Neurod6^; KO* mice compared with those of control mice (*n* = 4 animals per genotype; one-way ANOVA with Tukey’s multiple-comparison test). Scale bars as indicated in each panel. Individual data points were shown as dots with group mean ± S.E.M; * *p* < 0.05; ** *p* < 0.01; *** *p* < 0.001; n.s.: Not significant.

**Figure 2 ijms-22-08394-f002:**
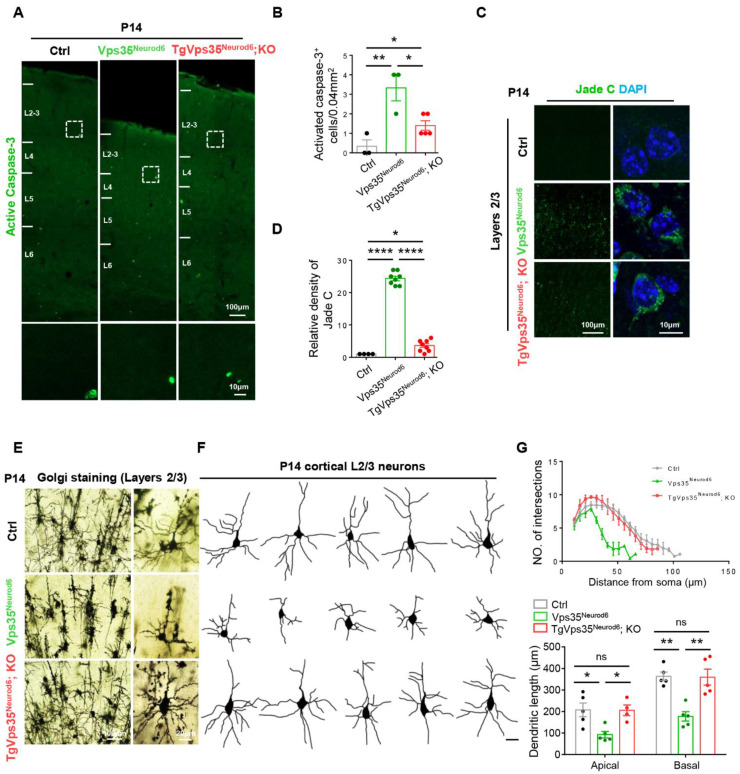
Vps35-mCherry expression rescues cell death and terminal differentiation deficits of *Vps35^Neurod6^* pups. (**A**) Representative images of active caspase-3 immunostaining analysis of neocortex at P14. (**B**) Quantification analysis of active caspase-3^+^ cells (*n* = 3 animals per genotype; one-way ANOVA with Tukey’s multiple-comparison test). (**C**) Representative images of Jade C staining. (**D**) Quantification analysis of Jade C^+^ cells (*n* = 4–8 views from three animals per genotype; one-way ANOVA with Tukey’s multiple-comparison test). (**E**) Representative images of a Golgi staining pyramidal neuron from the L2-3 projection neurons. (**F**) Tracing of representative L2-3 pyramidal neurons and assessed by Sholl analysis. (**G**) Quantification of dendritic complexity as well as apical and basal dendrites length (*n* = 5 neurons from three animals per genotype; one-way ANOVA with Tukey’s multiple-comparison test). Scale bars as indicated in each panel. Individual data points were shown as dots with group mean ± S.E.M; * *p* < 0.05; ** *p* < 0.01; **** *p* < 0.0001; n.s.: Not significant.

**Figure 3 ijms-22-08394-f003:**
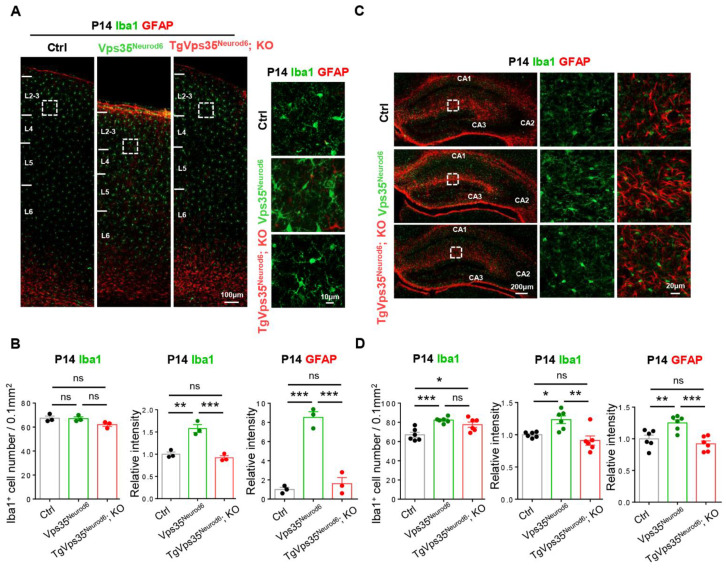
Vps35-mCherry expression rescues glial activation of *Vps35^Neurod6^* pups at P14. (**A**,**C**) Representative images of immunostaining analysis using indicated antibodies in P14 neocortical and hippocampal sections from mice with labelled genotype. Higher-magnification images of the boxed regions were shown in offside panels. (**B**,**D**) Quantification of immunofluorescence from A,C. (For A,B: *n* = 3 animals per genotype; for C,D: *n*= 6 views from three animals per genotype; one-way ANOVA with Tukey’s multiple-comparison test). Scale bars as indicated in each panel. Individual data points were shown as dots with group mean ± S.E.M; * *p* < 0.05; ** *p* < 0.01; *** *p* < 0.001; n.s.: Not significant.

**Figure 4 ijms-22-08394-f004:**
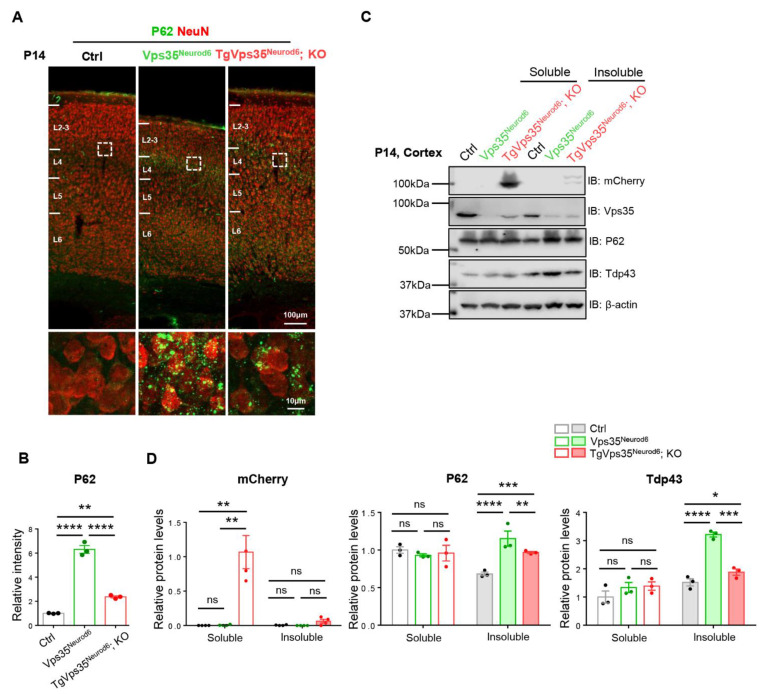
Partial recovery of FTD (frontotemporal dementia)-like neuropathology in *Vps35^Neurod6^* neocortex. (**A**) Representative images of co-immunostaining analysis with antibodies against P62 (green) and NeuN (red) of P14 cortical sections. (**B**) Quantification analysis of immunofluorescence of from A (*n* = 3 animals per genotype; one-way ANOVA with Tukey’s multiple-comparison test). (**C**) Western blot analyses of soluble (by 1% TritonX-100) and insoluble fractions (by 7 M Urea, 2 M Thiourea, 4% CHAPS) of homogenates of P14 control, *Vps35^Neurod6^* and *TgVps35^Neurod6^; KO* mice using indicated antibodies. (**D**) Quantification analysis of relative protein expression levels from C (*n* = 3 animals per genotype; one-way ANOVA with Tukey’s multiple-comparison test). Scale bars as indicated in each panel. Individual data points were shown as dots with group mean ± S.E.M; * *p* < 0.05; ** *p* < 0.01; *** *p* < 0.001; **** *p* < 0.0001; n.s.: Not significant.

**Figure 5 ijms-22-08394-f005:**
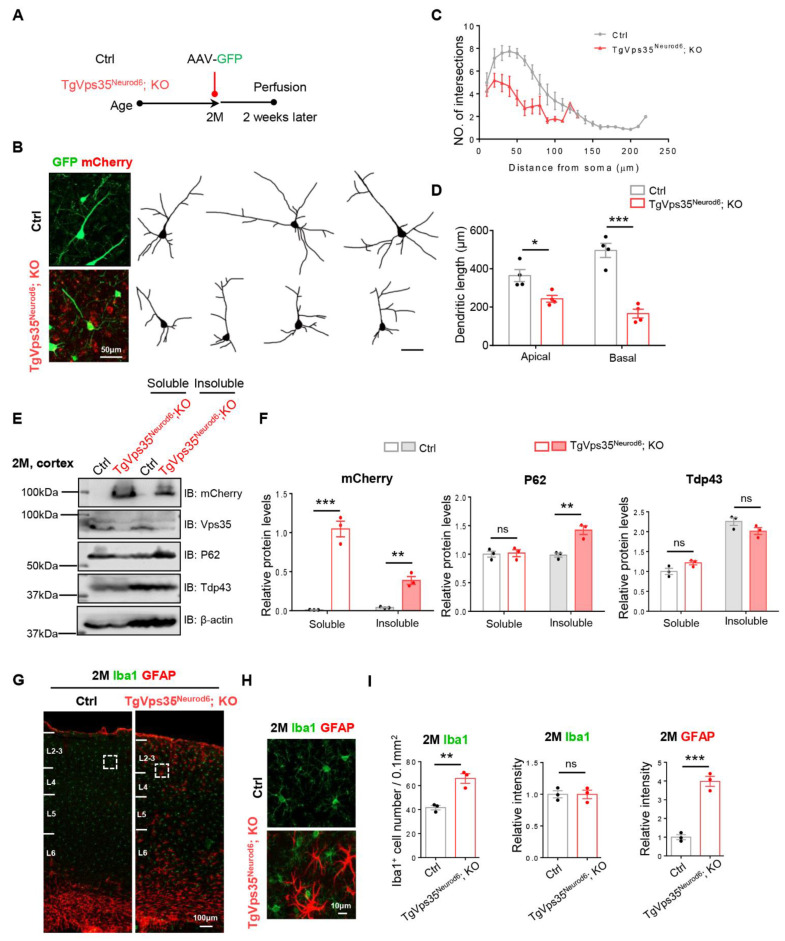
Terminal differentiation deficits, FTD (frontotemporal dementia)-like neuropathology, and increased GFAP^+^ reactive astrocytes and Iba1^+^ microglia in *TgVps35^Neurod6^; KO* cortex. (**A**) Time graph illustrating the process of AAV-Syn-GFP injection in control (*Vps35^f/f^*) and *TgVps35^Neurod6^; KO* mice. (**B**) Representative images and tracing of single neuron labelled with GFP from L2-3 of control and *TgVps35^Neurod6^; KO* mice. (**C**,**D**) Quantification shows decreased dendritic complexity by Sholl analysis and a reduction of apical and basal dendrites length (*n* = 4 neurons from three mice per genotype; two-tailed unpaired t-test). (**E**,**F**) Western blot analyses and quantification analysis of relative protein expression levels from E (*n* = 3 animals per genotype; two-tailed unpaired t-test). (**G**,**H**) Representative images of immunostaining analysis using indicated antibodies in 2-months-old neocortical sections from mice with labelled genotype. Higher-magnification images of the boxed regions were shown in offside panels. (**I**) Quantification of immunofluorescence from G,H. (*n* = 3 animals per genotype; two-tailed unpaired *t*-test). Scale bars as indicated in each panel. Individual data points were shown as dots with group mean ± S.E.M; * *p* < 0.05; ** *p* < 0.01; *** *p* < 0.001; n.s.: Not significant.

**Figure 6 ijms-22-08394-f006:**
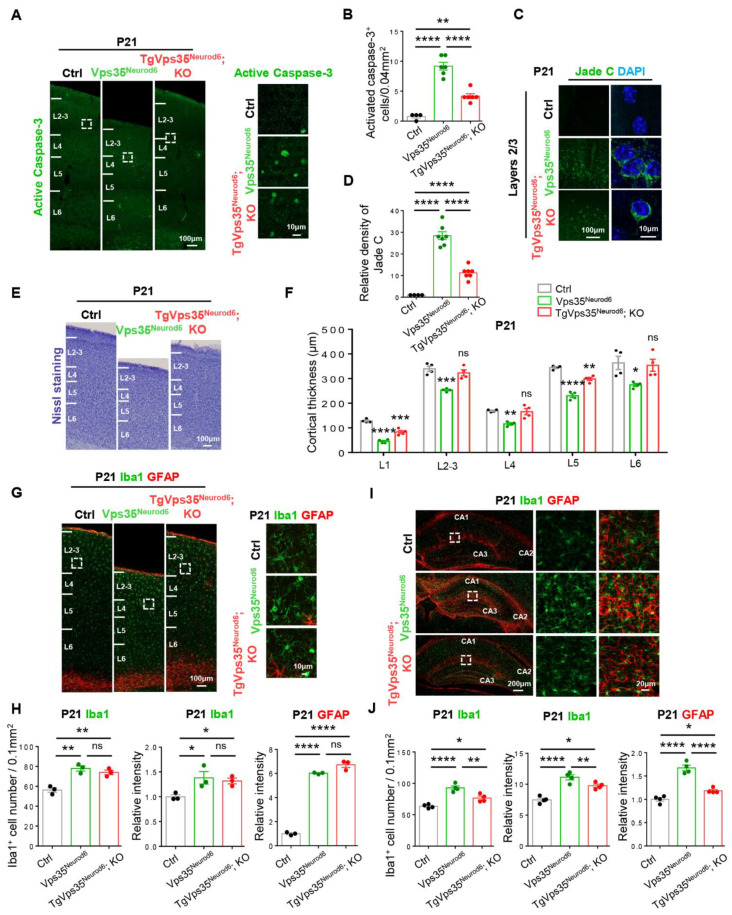
Cell death and glial activation in *TgVps35^Neurod6^; KO* mice at P21. (**A**) Representative images of active caspase-3 immunostaining analysis of neocortex at P21. (**B**) Quantification analysis of active caspase-3^+^ cells (*n* = 4–6 views from three animals per genotype; one-way ANOVA with Tukey’s multiple-comparison test). (**C**) Representative images of Jade C staining at P21. (**D**) Quantification analysis of Jade C^+^ cells (*n* = 4–7 views from three animals per genotype; one-way ANOVA with Tukey’s multiple-comparison test). (**E**) Representative Nissl staining of control, *Vps35^Neurod6^* and *TgVps35^Neurod6^; KO* mice at P21. (**F**) Quantification analysis of Nissl staining that revealed a reduced cortical thickness of *TgVps35^Neurod6^; KO* mice compared with those of control mice (*n* = 4 animals per genotype; one-way ANOVA with Tukey’s multiple-comparison test). (**G**,**I**) Representative images of immunostaining analysis using indicated antibodies in P21 neocortical and hippocampal sections from mice with labelled genotype. Higher-magnification images of the boxed regions were shown in offside panels. (**H**,**J**) Quantification of immunofluorescence from G,I. (For G,H, *n* = 3 animals per genotype; for I, J, *n* = 4 animals per genotype; one-way ANOVA with Tukey’s multiple-comparison test). Scale bars as indicated in each panel. Individual data points were shown as dots with group mean ± S.E.M; * *p* < 0.05; ** *p* < 0.01; *** *p* < 0.001; **** *p* < 0.0001; n.s.: Not significant.

**Figure 7 ijms-22-08394-f007:**
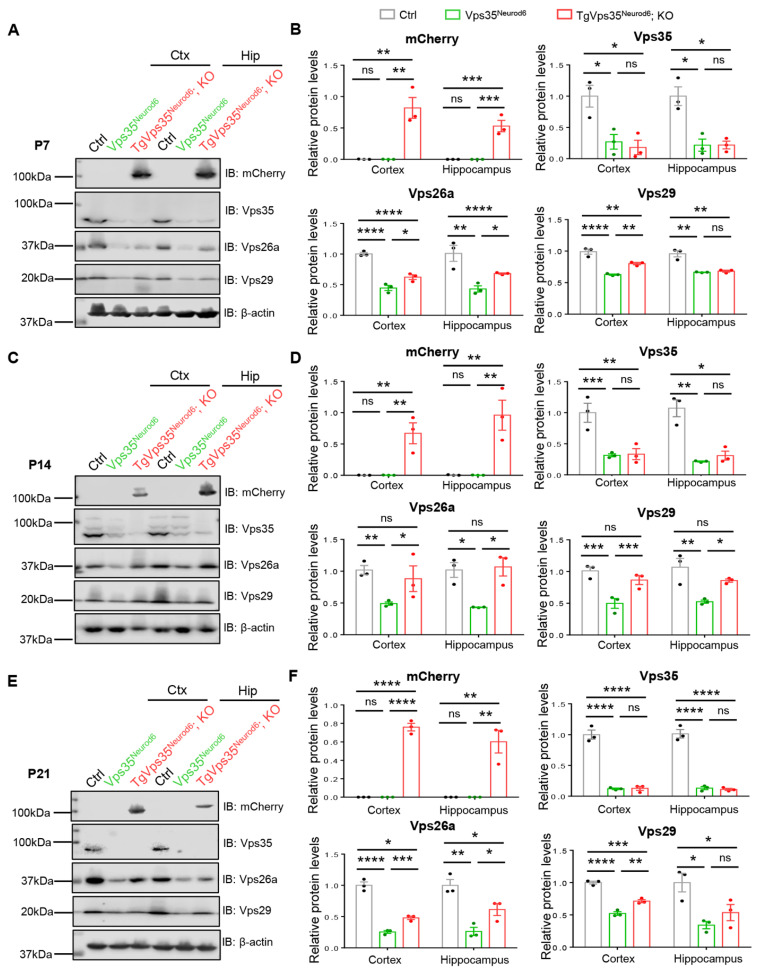
Vps35-mCherry expression modulates retromer complex. (**A**,**C**,**E**) Western blot analyses of homogenates of control, *Vps35^Neurod6^* and *TgVps35^Neurod6^; KO* brains at different ages using the indicated antibodies. (**B**,**D**,**F**) Quantification analysis of relative protein expression levels from A, C, and E (*n* = 3 animals per genotype; one-way ANOVA with Tukey’s multiple-comparison test). Individual data points were shown as dots with group mean ± S.E.M; * *p* < 0.05; ** *p* < 0.01; *** *p* < 0.001; **** *p* < 0.0001; n.s.: Not significant.

**Figure 8 ijms-22-08394-f008:**
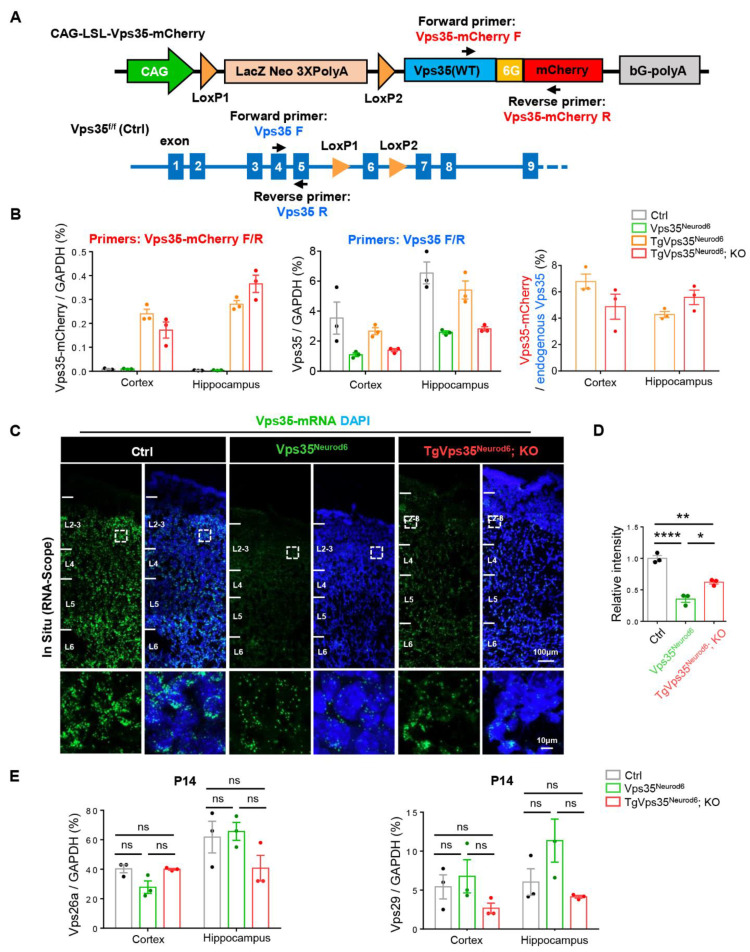
Low level of *Vps35-mCherry*’s expression. (**A**) Schematic graph illustrating the primers design for real-time RT-PCR (qRT-PCR) analysis. (**B**) Quantification of *Vps35-mCherry* and *Vps35* mRNA expression in control, *Vps35^Neurod6^*, *TgVps35^Neurod6^* and *TgVps35^Neurod6^; KO* brains (*n* = 3 animals per genotype). (**C**) Representative images of RNA scope analysis of *Vps35* mRNAs (green) in control, *Vps35^Neurod6^* and *TgVps35^Neurod6^; KO* cortex. The nuclei were stained with DAPI (blue). (**D**) Quantification analysis of data in (**C**) (*n* = 3 animals per genotype; one-way ANOVA with Tukey’s multiple-comparison test). (**E**) Quantification of *Vps26a* and *Vps29*’s mRNA levels (*n* = 3 animals per genotype; one-way ANOVA with Tukey’s multiple-comparison test). Scale bars as indicated in each panel. Individual data points were shown as dots with group mean ± S.E.M; * *p* < 0.05; ** *p* < 0.01; **** *p* < 0.0001; n.s.: Not significant.

## Data Availability

The data presented in this study are available in Appendix A.

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
