# Peer review of "Expression of Low Level of VPS35-mCherry Fusion Protein Diminishes Vps35 Depletion Induced Neuron Terminal Differentiation Deficits and Neurodegenerative Pathology, and Prevents Neonatal Death"

_ijms, 2021, doi:10.3390/ijms22168394_

Round 1

Reviewer 1 Report

The manuscript by Zhou and colleagues describes a partial and temporary rescue of cell damage and glia recruitment in the neuron-specific deletion of VPS35 by expressing VPS35-mCherry from a Cre-regulated transgene. The data is well presented and clearly quite some work has gone into the cross-breeding of these mouse-lines and the histological analysis. That said, this paper does not actually deliver new insight in how VPS35 depletion would lead to the studied changes. As the authors acknowledge in the discussion of their work, these results can be fully explained by poor expression of the transgene at different timepoints. This is a technical challenge of an exogenous expression model, which does not further our understanding of VPS35 function in the brain.

Other comments:

  • There seems to be a mismatch between the number of presented data points and the n numbers mentioned in the legend. It is not clear what each of these data points represents.
  • Fig 5 lacks a VPS35 null condition, thus no conclusions should be made on how the TgVPS35-mCherry compares to this situation. Furthermore, “re-occurance” is an overstatement apart from this issue. Animals are not studied over time but different animals are sacrificed at different time points, so “re-occurance” of a phenotype is impossible in this experimental design.
  • The expressed VPS35-mCherry construct is not detected in anti-VPS35 blots (Fig4, Fig7). This is a critical issue because only from such blots the relative protein expression of the transgene could be inferred. The explanation of the authors on page 16 (line 303-307) about antibody sensitivity is not satisfactory as longer exposures, different buffers or antibody concentrations can be used to solve this issue. The antibody is sensitive enough to pick up a band in the VPS35 knock out condition, on which the authors do not comment. In Fig8, the authors do present not present mRNA levels of endogenous VPS35, only as a ratio to transgene transcript which is little informative as control conditions and null mutant alone can not be included.
  • The title is an overstatement. No options for therapy of low level VPS35 expression are presented or supported (quite the opposite, neurodegeneration still occurs) and neuron terminal deficits upon VPS35 manipulation are not studied in this manuscript.

Author Response

Response to Reviewer 1 Comments

We thank Reviewer 1 for his/her comments that “The manuscript by Zhou and colleagues describes a partial and temporary rescue of cell damage and glia recruitment in the neuron-specific deletion of VPS35 by expressing VPS35- mCherry from a Cre-regulated transgene. The data is well presented and clearly quite some work has gone into the cross-breeding of these mouse-lines and the histological analysis.”

We also appreciate reviewer 1’s constructive comments. A point to point response to his/her comments is below.

Point 1: This paper does not actually deliver new insight in how VPS35 depletion would lead to the studied changes. As the authors acknowledge in the discussion of their work, these results can be fully explained by poor expression of the transgene at different timepoints. This is a technical challenge of an exogenous expression model, which does not further our understanding of VPS35 function in the brain.

Response: While we thank the reviewer for his/her comments, we believe that this paper has the following new insights into VPS35’s function in neurons. First, our results suggest that the low level of Vps35-mCherry fusion protein expression is sufficient to diminish the deficits in retromer complex and neuronal terminal differentiation, and to attenuate the increases in cell death and gliosis, at neonatal age (see revised Figs. 2-4, 6-7). These results not only verified neuronal VPS35’s age-dependent functions, but also implicate a potential therapeutic strategy (e.g., expressing low level of VPS35 in various ages by multiple injections might be beneficial or effective to prevent neurodegenerative pathology). We will further test this strategy in future experiments. Second, our results that after passive neonatal development, the mutant mice (TgVPS35-CKO) could survive to ages of >8 months old (Fig. 1D), even though they have neuro-degenerative pathology, suggesting that this mouse line may be a new animal model to investigate age-dependent retromer’s functions, neurodegenerative pathology, and behavior phenotypes. We have included these points in revised Discussion (see page 21).

Point 2: There seems to be a mismatch between the number of presented data points and the n number mentioned in the legend.

Response: Sorry for the mistake. The figure legends (Fig. 1-8) have been revised accordingly.

Point 3: Fig 5 lacks a VPS35 null condition, thus no conclusions should be made on how the TgVPS35-mCherry compares to this situation.

Response: Notice that Vps35Neurod6 conditional knock out mice undergo neonatal death (before P27) (Tang et al., 2020). Thus, at age of 2-months old (MO), TgVPS35-CKO, but not Vps35Neurod6, mice remain survived. We therefore compared TgVPS35-CKO’s phenotypes with those of littermate control mice (VPS35f/f). Although we could not conclude how much the transgene diminishes the deficits (which need to compare with the phenotypes in Vps35Neurod6 mice), we could reach the conclusion that the phenotypes detected in 2-MO TgVPS35-CKO, but not in their littermate control mice, were similar to those brain pathologies observed in P14-P21 Vps35Neurod6 mice, which are described in our previous publication (Tang et al., 2020).

Point 4:Re-occurrence” is an overstatement apart from this issue, animals are not studied over time, “Re-occurrence” of a phenotype is impossible in this experimental design.

Response: Agree! We have revised this statement (see page 12-16). 

Point 5: The expressed VPS35-mCherry construct is not detected in anti-VPS35 blots, longer exposures, different buffers or antibody concentrations can be used to solve this issue.

Response: We have made our efforts on this issue, including exposed the film for longer time, tested different buffers, and increased antibody concentrations as suggested. However, we failed to detect the larger molecular weight fusion protein band by the antibodies against VPS35 in tissue homogenates from the transgenic mouse line. We speculate that the following reasons may underlie such a negative effect. First, the anti-VPS35 antibody appears to be much less sensitive than that of mCherry antibody. As shown in Fig. S1B, the sensitivity of mCherry antibody appeared to be >10 fold more than that of anti-VPS35 in detecting the VPS35-mCherry fusion protein by Western blot. Second, the expression level of VPS35-mCherry is low, as shown in revised Fig. 8. These points have been included in revised Discussion (see page 20).   

Point 6: In Fig 8, the authors do not present mRNA levels of endogenous VPS35.

Response: The data of mRNA levels of endogenous VPS35 are included in revised Fig. 8.

Point 7: The title is an overstatement. No options for therapy of low level VPS35 expression are presented or supported and neuron terminal deficits upon VPS35 manipulation are not studied in this manuscript.

Response: Agree! Revised.

Reviewer 2 Report

Review of a manuscript “Therapeutic effect of low level of Vps35-mCherry fusion protein in Vps35 depletion induced neuron terminal differentiation deficits, neurodegenerative pathology, and neonatal death” by Yang Zhao and coauthors submitted to IJMS.

Alzheimer’s disease and other neurodegenerative disorders put an enormous pressure on healthcare system and represent a heavy problem for patients and their family members.  One of the component playing a role in neurodegeneration is a retromer complex, an endosomal protein sorting machinery. More specifically, the authors investigated whether VPS35 – a key component of retromer complex may be considered as a potential therapeutic target for neurodegenerative diseases. This research and the results presented in the manuscript might be important and interesting for the readers of IMGS. The following corrections should be made.

Abstract

Lanes 23-26:”Further  studies  revealed  that  the  Vps35-mCherry transgene’s expression was low, ~5-7% of Vps35’s mRNA of control mice; and such a low level of VPS35-mCherry could restore the amount of other retromer components (Vps26a and Vps29) at the neonatal age (P14); and the neurodegenerative pathology re-occurred in the survived adult TgVps35-mCherry; Vps35Neurod6mice.”

This sentence is too long and hard to read and understand. It should be split into three as follows:

”Further studies revealed that the Vps35-23mCherry transgene’s expression was low, and the level of Vps35’s mRNA comprised only ~5-7% of Vps35’s mRNA of control mice. Such low level of VPS35-mCherry could restore the amount of other retromer components (Vps26a and Vps29) at the neonatal age (P14). Importantly, the neurodegenerative pathology re-occurred in the survived adult TgVps35-mCherry Vps35Neurod6mice”.

Introduction

Lanes 83-85:”Here, we  provide  evidence  for  expression of  VPS35-mCherry  fusion  protein  in Vps35 Neurod6 mice could prevent neonatal death of Vps35 Neurod6mice, and attenuates the dendritic morphogenesis deficit and gliosis, but not increases of P62 and Tdp43, specifically at the neonatal age”. The sense of this sentence is unclear. It should be corrected as follows: ”Here, we provide evidence that expression of VPS35-mCherry fusion protein in Vps35Neurod6mice prevents neonatal death of Vps35Neurod6mice. Moreover, it attenuates the dendritic morphogenesis deficit and gliosis, but do not increase P62 and Tdp43, specifically at the neonatal age”.

Results

Figure 1: (G) Representative Nissl stains of control, Vps35Neurod6 and TgVps35Neurod6; KO mice at P14. The quality of the figure should be improved. Nissl stains should be written as Nissl staining.

Figure 4. The authors should provide a larger image for better quality of image.

Figure 6, A and G – again better quality of images should be provided, since it is hard to see the details.

Lanes 290 :” 2.7 Age-dependent restore of retromer complex…” Do the authors mean “2.7 Age-dependent restoration of retromer complex”?

Discussion

Lanes 367-368: “Studies of genetically manipulated mouse models have proven animal models to be valuable tools for our understanding of pathological mechanisms of human disease.”

 This is a clumsy sentence which should be corrected as follows: ”Genetically manipulated mouse models have proven to be valuable tools for our understanding of pathological mechanisms of human diseases.”

Author Response

Response to Reviewer 2 Comments

We thank Reviewer 2 for his/her comments that “Alzheimer’s disease and other neurodegenerative disorders put an enormous pressure on healthcare system and represent a heavy problem for patients and their family members. One of the component playing a role in neurodegeneration is a retromer complex, an endosomal protein sorting machinery. More specifically, the authors investigated whether VPS35 – a key component of retromer complex may be considered as a potential therapeutic target for neurodegenerative diseases. This research and the results presented in the manuscript might be important and interesting for the readers of IMGS”.

We have taken Reviewer 2’s suggestions and made the following corrections.

Point 1: Abstract: Lanes 23-26: The sentence is too long and hard to read and understand. It should be split into three as follows: “Further studies revealed that the Vps35-23mCherry transgene’s expression was low, and the level of Vps35’s mRNA comprised only ~5-7% of Vps35’s mRNA of control mice. Such low level of VPS35-mCherry could restore the amount of other retromer components (Vps26a and Vps29) at the neonatal age (P14). Importantly, the neurodegenerative pathology re-occurred in the survived adult TgVps35-mCherry Vps35Neurod6 mice.

Response: Revised as suggested (see page 1)

Point 2: Lane 83-85: The sense of this sentence is unclear. It should be corrected as follows: “Here, we provide evidence that expression of VPS35-mCherry fusion protein in Vps35Neurod6mice prevents neonatal death of Vps35Neurod6mice. Moreover, it attenuates the dendritic morphogenesis deficit and gliosis, but do not increase P62 and Tdp43, specifically at the neonatal age.”

Response: Thanks for the suggestion. Revised (see page 3).

Point 3: Figure 1: (G) Representative Nissl stains of control, Vps35Neurod6 and TgVps35Neurod6; KO mice at P14. The quality of the figure should be improved. Nissl stains should be written as Nissl staining.

Response: Revised as suggested, and the improved images are included in revised Figure 1G.

Point 4: Figure 4. The authors should provide a larger image for better quality of image.

Response: Agree! An enlarged image with a better quality is now included in revised Fig. 4.

Point 5: Figure 6, A and G – again better quality of images should be provided, since it is hard to see the details.

Response: Yes, we have improved the quality of the images in Figure 6.

Point 6: Lanes 290: “Age-dependent restore of retromer complex...” Do the authors mean “2.7 Age-dependent restoration of retromer complex?”

Response: Yes, revised as suggested (see page 16).

Point 7: Lanes 367-368: This is a clumsy sentence which should be corrected as follows: “Genetically manipulated mouse models have proven to be valuable tools for our understanding of pathological mechanisms of human diseases.”

Response: Thanks for the suggestions. Revised as suggested (see page 20).